# Mortality Trends Related to Bladder Cancer in Spain, 1999–2018

**DOI:** 10.3390/jcm11040930

**Published:** 2022-02-10

**Authors:** Pau Sarrio-Sanz, Laura Martinez-Cayuelas, Vicente Francisco Gil-Guillen, José Antonio Quesada, Luis Gomez-Perez

**Affiliations:** 1Urology Department, University Hospital of San Juan de Alicante, San Juan de Alicante, 03550 Alicante, Spain; pausarrio@gmail.com (P.S.-S.); martinezcayuelaslaura@gmail.com (L.M.-C.); luisgope@gmail.com (L.G.-P.); 2Department of Clinical Medicine, Miguel Hernández University, San Juan de Alicante, 03550 Alicante, Spain; vte.gil@gmail.com

**Keywords:** bladder cancer, trends, cancer-specific mortality, premature mortality

## Abstract

Bladder cancer (BC) is an important cause of premature mortality (PM, <75 years). Spain has one of the highest BC mortality rates in Europe. The objective of this study was to analyse BC mortality trends between 1999 and 2018 in Spain. The study was based on data from the National Institute of Statistics (Instituto Nacional de Estadística—INE). Age-adjusted mortality rates (AAMRs) were calculated by sex and age group. A trend analysis was performed using Joinpoint regression models and years of potential life lost (YPLL). Mortality in men resulting from BC decreased in all age groups studied. This was not observed in women, for whom mortality only decreased in the ≥75 age group. Deaths due to BC occurred prematurely in 38.6% of men and in 23.8% of women, which indicated a greater impact on YPLL in men compared to women. Over the last 20 years, there has been a significant decrease in BC mortality rate, except in women under 75 years of age. Despite this temporal trend of decreasing mortality, BC continues to have a significant impact on YPLL, mainly in men. Given this context, it is important to direct more resources towards prevention and early diagnosis strategies to correct this situation.

## 1. Introduction

Bladder cancer (BC) is a disease in which tumours affect the urothelium of the bladder. Urothelial carcinoma is the most common histological variant of bladder tumours. Bladder cancers are classified as non-muscle invasive (superficial) or muscle invasive (infiltrative) (MIBC) tumours based on whether or not they affect the muscular layer of the bladder. Non-muscle invasive tumours account for 75% of cases, with a high recurrence rate but a low rate of progression and mortality [1,2]. Muscle-invasive patients have a worse prognosis, with a 5-year cancer-specific survival rate between 23.5% and 65% [1,3,4,5].

The main risk factor is tobacco (both in active and passive smokers), which is involved in more than 50% of cases [1,2,3]. In Spain, anti-smoking legislation (Law 28/2005 and its modification by Law 42/2010) prohibits smoking in any type of closed space for collective use. 

Tobacco use in Spain has decreased (improving from 24th place in 2004 to 8th place in 2016 in the Tobacco Control Scale Ranking) [6], although 22–29% of Spanish adults are smokers and 19–25% are former smokers [7,8]. Exposure rates of passive smokers have also decreased from 75% in 2004–2005 to 56.7% in 2011–2012 [6]. The proportion of active male smokers has been decreasing progressively since data have been available; however, there was an increase in female smokers in the 1980s and 1990s. From 2002 to 2006, the proportion of female smokers began to slowly decrease [7,9].

Occupational exposure is responsible for 5–6% of the attributable risks of developing BC, resulting from exposure to chemicals on the part of workers in contact with dyes, paints, hydrocarbons, glues, rubber or aluminium, among other substances [3]. Exposure to these carcinogens and lack of worker protection measures could explain variations in incidences between different geographic areas [10]. The Occupational Risk Prevention Law 31/1995 was enacted in Spain in 1995 and subsequently amended in 2014.

In terms of BC treatment, in recent decades, neoadjuvant chemotherapy (NAC) has been implemented for radical cystectomy, new drugs have been developed and radical cystectomies have been carried out on older patients [1,11]. The use of NAC for radical cystectomy was proposed as far back as 1980. This method has had an impact on improving cancer-specific survival rates as well as overall survival rates [12], and there has been an increase in its use since 2010 [13,14]. In Spain, no studies of trends in the use of NAC in MIBC exist; however, data can be extrapolated from different published series. These data indicate that between 3.1 and 4.3% [15,16] of patients received NAC before 2010, with a percentage increase in its current use varying between 29.7% and 60.7% [16,17,18]. The indication for surgery in geriatric patients have been increasingly positive as surgical experience and life expectancy have improved [19], with cystectomy becoming established as a safe procedure in selected elderly patients [20,21].

BC has a very prevalent pathology—it was the fifth most common cancer in Europe in 2020 [22,23]. In Spain, it is also the fifth most prevalent form of cancer, affecting 150 per 100,000 inhabitants, with 75% of these being men [24].

The age-adjusted rate (AAR) of BC incidence per 100,000 in Spain is high. The AAR in men (39/100,000) is above the European average (29.1/100,000) and is among the highest, with rates similar to those in Italy, Germany and Belgium. In women it is estimated to be 5.5/100,000, which is below the European average (6.1/100,000) and is similar to rates found in countries such as Finland, the United Kingdom and Italy [22,25].

In terms of mortality, BC is the second highest cause of death from urological tumours (behind prostate cancer) and is responsible for at least 165,000 deaths/year worldwide [23,26]. In Spain, the crude mortality rate is 12/100,000, with significant differences in terms of gender comparison [27]. In women, the age-adjusted mortality rate (AAMR) is 1.08/100,000 (which is the European average) compared to 8.1/100,000 in men, which is one of the highest rates in Europe [24,28,29].

On the other hand, BC is an important cause of premature mortality (PM), which is defined as any death before the average life expectancy at birth [30,31]; in Spain, the average life expectancy is 75 years [32]. One of the methods for measuring premature mortality is the years of potential life lost (YPLL). In the United States alone, it is estimated that between 2002 and 2006, 254,540 YPLL were lost due to BC [33].

Increases in mortality due to BC have been observed in Spain since at least 1955, with the highest increase being in men in the years after 1970 [34,35,36]. From 1993 to 2007, rates of BC increased for both men and women; however, mortality rates for men remained stable [24,28,37,38], while a slight decrease was recorded for women [38,39]. The projections derived from these data [27] predicted a decrease in rates of BC incidence and mortality for men and an increase in both rates for women. However, these projections have not been analysed, and there have been no studies that investigated BC mortality trends in Spain over the last 15 years.

Understanding the current PM situation resulting from BC is very important. Although there are some areas of debate [40,41], BC is generally considered a potentially avoidable cause of death [42,43]. The YPLL estimate and PM analysis alongside mortality trends from the age of 75 could underpin and orientate health strategies as well as specific prevention initiatives, along with early diagnostic protocols. 

The objective of this study is to analyse temporal PM trends and rates among those above the age of 75 due to BC in Spain according to gender, as well as the YPLL between 1999 and 2018.

## 2. Materials and Methods

A study of temporal mortality trends in Spain due to BC was carried out for the years between 1999 and 2018.

All patients who died from BC in Spain were included for the study period, and data were obtained from registries in Spain’s National Institute of Statistics (Instituto Nacional de Estadística—INE) database [19], which include mortality data and causes of death in the Spanish population as a whole. The identification of BC as a cause of death was established using the C67 code as set out in the tenth review of the International Classification of Diseases (ICD10).

All causes of death were analysed, including specific mortality due to BC and proportional mortality (percentage of deaths from BC compared to the total), after stratification of the sample by sex and age (<75 years or ≥75 years). Age stratification allowed for PM analysis (defined as happening at <75 years). AAMRs, YPLL and mortality rate trends were calculated.

AAMR is an estimate of the risk of death in a given population and allows for control of the effect of population age distribution for mortalities resulting from a specific cause. Calculations for this study were carried out using a direct method with respect to a reference population from the European standard population of 2013 published by Eurostat [44]. AAMR was calculated using five-year age bands for each year and separating each band by gender with a 95% confidence interval (95% CI).

The YPLL was calculated for each year and for each age group separated by sex.

Mortality trends were analysed by using Joinpoint regression models, and the annual percentage change (APC) in each segment and the average annual percentage change (AAPC) for the entire period were estimated with a 95% CI. The selection of the models was carried out using a permutation test, adjusting for autocorrelated errors based on the data [45]. 

Statistical analysis was carried out using the SPSS V.26 statistical package [46] to calculate rates, and the Joinpoint Regression Program V.4.8.0.1 [47] was used to determine statistical regression and mortality trends.

This project has been accepted by our institution’s Ethics and Integrity in the Research Committee (code 210713141516).

## 3. Results

During the study period, 4,382,650 male deaths and 3,930,815 female deaths were recorded in Spain; of these, 81,282 (1.86%) and 17,500 (0.45%), respectively, were a result of BC (Table 1 and Table 2).

Mortality analysis shows that proportional mortality in males due to BC as compared to total mortality was constant in both age groups (Table 1). PM AAMRs in men fell during the period under study (AAPC of −2.8). In 2012, there was a significant trend change from an annual APC of −1.4% to −5.9%. Despite the observed mortality trends, 28,953 deaths in men (38.6%) occurred prematurely, amounting to 248,921 YPLL (Table 3, Figure 1). In the ≥75 years age group, male mortality increased up to 2012 (APC + 0.46) until a trend change occurred for that year with an APC of −3.8% (Table 3, Figure 1).

Proportional mortality in women with BC compared to total mortality saw a slight increase in the group below 75 years of age (0.4% to 0.51%). However, statistical adjustment by age shows that PM remained constant (APC −0.10 (−0.50, 0.30 IC)). Despite this, 3996 deaths occurred prematurely (23.8% of deaths resulting from BC), resulting in 38,482 YPLL. After the age of 75, mortality does not change significantly until 2013, when a significant decrease was observed with an APC of −4.1% per year. This decrease was comparatively greater than that observed in men (Table 3, Figure 1).

## 4. Discussion

Analysis of the data showed a decrease in the mortality rate due to BC in Spain between 1999 and 2018, except in women under 75 years old.

Similar decreases in BC mortality have been observed in most advanced economies across different continents in both men and women, while in countries such as Cuba, China, Slovenia, Croatia and Bulgaria, this has not been the trend [22,25].

Compared to other countries that showed decreases in BC mortality rates, the annual percentage changes (APC) for Spain from 2012 onward (at between −3.8 and −5.9) present some of the bigest declining trends in mortality related to BC identified worldwide [24,28]. Despite these declines in mortality rates, BC continues to have a significant impact on YPLL in Spain, particularly in men.

According to data from the Spanish Cancer Registry Network (Red Española de Registros de Cáncer—REDECAN) [25,48,49], incidences of BC in Spain have increased in men and even doubled in women over the last 20 years. Therefore, we believe that decreases in mortality cannot be attributed to declining numbers of cases. Demographic variations, the natural history of the disease (and exposure to risk factors), health services and changes in health information systems are probable explanations for the observed decreases in mortality rates.

Of the four groups of factors that may have influenced trends in BC mortality rates, there are two that are modifiable: exposure to risk factors and quality of health services. On the other hand, the impact of demographic variations and health information systems may restrict the interpretation of the results. These aspects are developed further below.

### 4.1. Exposure to Risk Factors

The introduction of the Anti-Smoking Law (Law 28/2005) and its subsequent amendment (Law 42/2010), as well as the consequent decrease in smoking in Spain, could explain the improvements in mortality figures. According to the data analysed, changes in mortality trends related to BC can be observed in men and women aged over 75 at 2 and 3 years, respectively, following the enactment of the law. Clearly, the variable latency period between exposure to tobacco, subsequent development of BC and mortality from the disease make it difficult to assess the direct impact of the Anti-Smoking Law. In addition, this study was not designed to evaluate the impact of the Anti-Smoking Law on changes in smoking habits.

It is difficult to know the true impact of the Law for the Prevention of Risk at Work and also to know the true impact of the different subsequent modifications to this law, as these have not been evaluated. Additionally, it is difficult to attribute occupational exposure to carcinogens as the sole cause of BC [3,10]. Although the incidence of BC related to occupational exposure increased in the decade between 2000 and 2010, mortality risk declined for both sexes [50]. Furthermore, a reduction in risk over time has been observed in several occupations, such as for drivers and for mechanics [50,51].

Other risk factors, such as infection by Schistosoma haematobium [1], are rare in Spain; therefore, these are not considered to have influenced the significant variations observed in the numbers of deaths.

### 4.2. Health Services: Diagnosis and Treatment

BC is considered an avoidable cause of death, both by primary prevention and by early detection and treatment, as well as by improvements in treatments and medical care [42].

BC prognoses are dependent on their stages and their grades. Delays in diagnoses and treatments affect overall survival. BC diagnoses are initially made based on symptoms, and in this context, primary care training is very important, along with advances in radiological tests (and having access to these).

Based on these findings, we believe that a reorientation of health systems towards prevention, as well as an establishment of rapid diagnostic and treatment circuits, could be expected to contribute to a reduction in BC mortality rates and the disease’s impact on YPLL without incurring excessive additional costs.

The risk of recurrence is high for non-muscle-invasive BC patients; hence, the use of biomarkers, cystoscopy surveillance and intravesical treatments can reduce its progression and, thus, its mortality [52].

New treatments such as PD-1/PD-L1 inhibitors have shown to decrease mortality in patients with metastatic BC. However, their impact on the mortality trends analysed here should not be considered significant, as recommendations for their use are relatively recent [1]. More time is probably needed to observe changes in mortality resulting from these treatments.

The extension of the indications for radical cystectomy to elderly patients [20,21] and the progressive incorporation of neoadjuvant chemotherapy [13,14] into treatment protocols in Spain may have contributed to the decrease in mortality observed in the trend analysis, particularly in the group aged 75 and older. For this reason, it is important to continue implementing bladder-sparing treatments and systemic treatments and to perform cystectomies in fit patients regardless of their age [53]. All of these treatments contribute to reducing bladder cancer-specific mortality.

### 4.3. Demographic Variations and Impact According to Sex

According to the data analysed, PM has not decreased in the women’s group under the age of 75. The mortality rate in this group is very low (age adjusted mortality rate of 1–1.1 per 100,000 inhabitants), which makes demonstrating a decrease in mortality very complicated in contrast to the situation involving men or women over the age of 75. Other conditioning factors, such as increased incidences in this group or social and health aspects that confer worse prognoses, must also be considered.

As women present more advanced diseases and have lower survival rates, gender is considered an independent risk factor for BC mortality [54]. Women with BC are considered to be at excess risk of mortality compared to men [55,56,57]. This can be explained by delays in diagnoses; for example, symptoms such as microhematuria or irritative voiding symptoms may be initially misidentified as benign pathologies (usually urinary infections), thus delaying diagnoses of BC [57].

On the other hand, changes in lifestyle and environmental exposure may increase BC incidences in women. An increase in the incidence rate of BC related to occupational exposure has been observed in women compared to men, which may contribute to the fact that there has not been a decrease in the mortality rate in younger women [50]. The increase in female smokers in the 1980s and 1990s (especially in young women) could be responsible for the rise in incidences and the consequent absence of declines in mortality [7,9]. If the BC incidence continues to increase, maybe in the next few years, there will be an increase in mortality in women.

In the literature, a decrease in mortality has been described in European women [24], but a stratified analysis by age allows us to affirm that premature mortality in women has not decreased, and this is probably the group that can benefit the most from specific interventions.

New studies are needed in order to analyse whether delayed diagnosis is a factor and whether there have been changes in levels of exposure to risk factors based on gender; moreover, there is a need to plan specific campaigns for early diagnosis and prevention of smoking in the population group of women under the age of 75.

### 4.4. Strengths and Limitations

To our knowledge, this is the first study to analyse BC mortality trends in Spain in the last 15 years. The length of the period under study and the methodology used in the analysis underpin the reliability of the conclusions, accurately establishing whether the data observed in mortality due to BC were random or not.

Despite the fact that the population database provided by the INE is an official and objective source with standardised coding methods, a possible variation in coding the cause of death due to BC could be a limitation when interpreting the results. In order to try to reduce the scope of this possible bias, data were selected following the implementation of the ICD-10 coding system in 1999. On the other hand, despite the reliability of the data obtained, a possible limitation of this study could be the absence of individual variables. This makes it impossible to identify individual factors that may be associated with changes in mortality detected, meaning that hypotheses can be generated but not confirmed. More research is needed to confirm the hypotheses developed in the present study.

Another limitation of the study relates to the possible impacts of demographic factors. Migratory flows or an ageing population may impact the variation of mortality trends observed; moreover, these variations may be a result of other modifiable health determinants. In order to reduce possibilities of these biases, AAMRs adjusted to the European standard population were used in addition to crude mortality rates.

### 4.5. Application to Clinical Practice

Based on the analysis of the data obtained, we consider it possible that the BC mortality rate will continue to improve. A reduction in the mortality rate and YPLL could be achieved by directing efforts towards smoking prevention and health education in people under 75 years of age. As far as early diagnosis is concerned, specific strategies are needed, such as training in early BC detection for primary care physicians and improvements in referral protocols for both men and women.

## 5. Conclusions

The last 20 years observed a decrease in BC mortality rates, except in women under 75 years of age. Despite this temporal trend towards decreasing mortality, BC continues to have a significant impact on YPLL, especially in men. In order to address this, resources should be allocated towards targeted prevention and early diagnosis strategies.

## Figures and Tables

**Figure 1 jcm-11-00930-f001:**
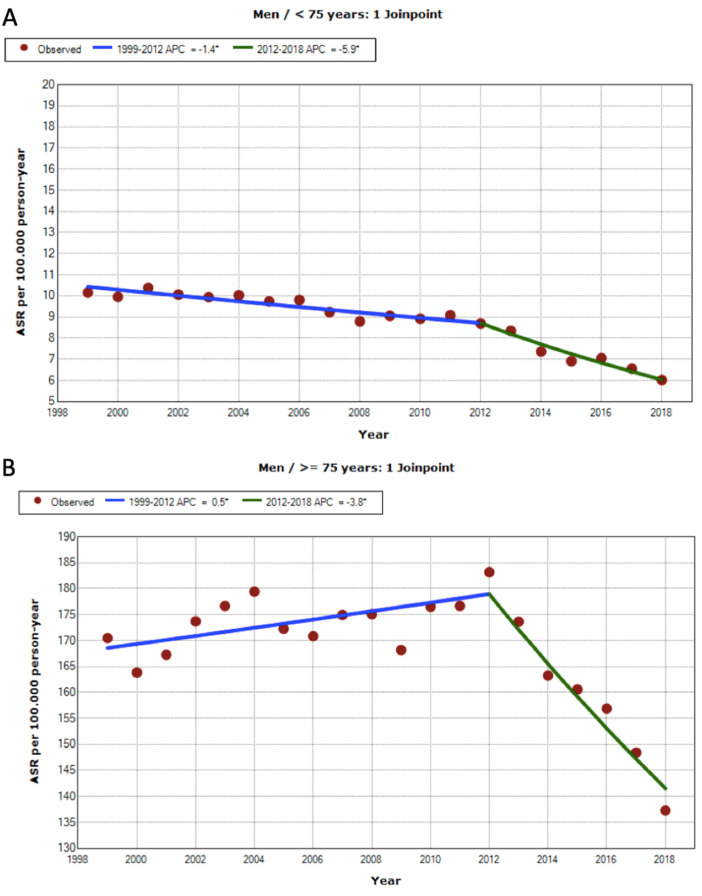
Premature mortality trends due to BC for the period 1999-2018 in Spain: (**A**) men < 75 years old (Final Selected Model = 1 Jointpoint); (**B**) men ≥ 75 years old (Final Selected Model = 1 Jointpoint); (**C**) women < 75 years old (Final Selected Model = 0 Jointpoint); and (**D**) women ≥ 75 years old (Final Selected Model = 1 Jointpoint). * Indicates that the Annual Percent Change (APC) is significantly different from zero at the alpha = 0.05 level.

**Table 1 jcm-11-00930-t001:** Total mortality, mortality due to BC, proportional mortality, AAMRs related to BC per 100,000 inhabitants and YPLL by age group in men in Spain during the period 1999–2018. TM: total mortality; BCM: bladder cancer mortality; AAMR: age-adjusted mortality rate per 100,000 inhabitants (direct method, European standard population of 2013); 95% CI: 95% Confidence Interval; YPLL: years of potential life lost.

<75 years							≥75 years				
YEAR	TM	BCM	BCM/TM (%)	AAMR	95% CI	YPLL	MR	BCM	BCM/TM (%)	AAMR	95% CI
1999	95944	1527	1.59	10.2	(9.6–10.7)	12629	99311	1702	1.71	170.5	(162.1–178.9)
2000	92934	1501	1.62	9.9	(9.4–10.5)	12822	96534	1692	1.75	163.9	(155.8–171.9)
2001	91535	1593	1.74	10.4	(9.9–10.9)	13661	98179	1781	1.81	167.2	(159.2–175.3)
2002	91100	1564	1.72	10.1	(9.6–10.6)	13148	102169	1929	1.89	173.7	(165.7–181.8)
2003	91646	1559	1.70	9.9	(9.4–10.4)	13728	108251	2009	1.86	176.7	(168.6–184.7)
2004	88397	1585	1.79	10.0	(9.5–10.5)	13645	106531	2127	2.00	179.4	(171.5–187.4)
2005	88825	1550	1.75	9.7	(9.2–10.2)	13550	112944	2123	1.88	172.3	(164.6–179.9)
2006	85326	1589	1.86	9.8	(9.3–10.3)	13883	108828	2153	1.98	170.8	(163.3–178.4)
2007	85680	1502	1.75	9.2	(8.8–9.7)	13289	115456	2253	1.95	174.9	(167.3–182.5)
2008	82954	1444	1.74	8.8	(8.3–9.2)	13178	116693	2383	2.04	175.1	(167.7–182.5)
2009	80691	1489	1.85	9.1	(8.6–9.5)	13038	118404	2365	2.00	168.2	(161.0–175.3)
2010	78521	1471	1.87	8.9	(8.5–9.4)	13032	119600	2562	2.14	176.5	(169.3–183.7)
2011	77541	1528	1.97	9.1	(8.6–9.5)	13751	122313	2626	2.15	176.7	(169.6–183.8)
2012	77101	1470	1.91	8.7	(8.2–9.1)	13255	128819	2850	2.21	183.2	(176.2–190.2)
2013	75165	1417	1.89	8.4	(7.9–8.8)	11824	124669	2774	2.23	173.6	(167.0–180.3)
2014	75813	1278	1.69	7.4	(7.0–7.8)	10576	125758	2621	2.08	163.3	(156.8–169.7)
2015	77543	1246	1.61	6.9	(6.5–7.3)	10587	135766	2614	1.93	160.6	(154.3–166.9)
2016	77183	1278	1.66	7.0	(6.7–7.4)	10271	131810	2640	2.00	156.9	(150.7–163.0)
2017	77798	1215	1.56	6.5	(6.2–6.9)	9795	136438	2517	1.84	148.4	(142.5–154.4)
2018	78934	1147	1.45	6.0	(5.7–6.4)	9259	137508	2367	1.72	137.3	(131.7–142.9)

**Table 2 jcm-11-00930-t002:** Total mortality, mortality due to BC, proportional mortality, AAMRs related to BC per 100,000 inhabitants and YPLL by age group in women in Spain during the period 1999–2018. TM: total mortality; BCM: bladder cancer mortality; AAMR: age-adjusted mortality rate per 100,000 inhabitants (direct method, European standard population of 2013); 95% CI: 95% Confidence Interval; YPLL: years of potential life lost.

<75 years							≥75 years				
YEAR	TM	BCM	BCM/TM (%)	AAMR	95% CI	YPLL	MR	BCM	BCM/TM (%)	AAMR	95% CI
1999	44559	178	0.40	1.0	(0.9–1.2)	1406	131288	514	0.39	29.1	(26.5–31.6)
2000	43433	190	0.44	1.1	(0.9–1.3)	1595	127490	531	0.42	29.2	(26.7–31.7)
2001	42733	204	0.48	1.1	(1.0–1.3)	1748	127684	531	0.42	27.7	(25.4–30.1)
2002	42103	174	0.41	1.0	(0.8–1.1)	1303	133246	530	0.40	26.9	(24.6–29.1)
2003	42763	189	0.44	1.0	(0.9–1.2)	1848	142168	536	0.38	26.5	(24.3–28.8)
2004	41014	185	0.45	1.0	(0.9–1.2)	1385	135992	599	0.44	28.8	(26.5–31.1)
2005	40883	199	0.49	1.1	(0.9–1.2)	1803	144703	558	0.39	26.0	(23.9–28.2)
2006	39255	194	0.49	1.1	(0.9–1.2)	1788	138069	590	0.43	26.6	(24.4–28.7)
2007	39534	202	0.51	1.1	(0.9–1.2)	2119	144691	614	0.42	27.0	(24.8–29.1)
2008	38853	200	0.51	1.1	(0.9–1.2)	2020	147824	677	0.46	28.7	(26.5–30.9)
2009	37876	196	0.52	1.1	(0.9–1.2)	1907	147962	682	0.46	27.9	(25.8–30.0)
2010	36687	213	0.58	1.1	(1.0–1.3)	2131	147239	690	0.47	27.1	(25.1–29.2)
2011	36693	214	0.58	1.1	(1.0–1.3)	2468	151364	717	0.47	27.3	(25.3–29.4)
2012	36169	202	0.56	1.1	(0.9–1.2)	2049	160861	731	0.45	27.2	(25.2–29.1)
2013	36247	212	0.58	1.1	(1.0–1.3)	2154	154338	750	0.49	27.2	(25.3–29.2)
2014	36737	168	0.46	0.9	(0.7–1.0)	1701	157522	733	0.47	25.9	(24.0–27.8)
2015	37512	231	0.62	1.1	(1.0–1.3)	2592	171747	708	0.41	24.4	(22.6–26.2)
2016	37512	218	0.58	1.1	(0.9–1.2)	2221	164106	725	0.44	24.7	(22.8–26.5)
2017	38715	229	0.59	1.1	(1.0–1.2)	2283	171572	659	0.38	22.0	(20.3–23.8)
2018	38831	198	0.51	0.9	(0.8–1.1)	1961	172448	688	0.40	22.4	(20.7–24.2)

**Table 3 jcm-11-00930-t003:** Annual percentage change (APC) and average annual percentage change (AAPC) in mortality due to BC estimated by Joinpoint regression in Spain by gender during the period 1999–2018. * *p* < 0.05; APC: annual percentage change for the corresponding segment; AAPC: annual average percentage change for the entire period; 95% CI: 95% confidence interval.

		Segment 1		Segment 2		Total
	Age	Period	APC (95% CI)	Period	APC (95% CI)	AAPC (95% CI)
			*p* Value		*p* Value	*p* Value
Men	< 75	1999–2012	−1.38 (−1.78; −0.97)	2012–2018	−5.95 (−7,37; −4,51)	−2.84 (−3.34; −2.34)
			<0.001 *		<0,001 *	<0.001 *
	≥ 75	1999–2012	0.46 (0.11; 0.82)	2012–2018	−3.83 (−4.83; −2.82)	−0.91 (−1.29 −0.54)
			0.015 *		<0.001 *	<0.001 *
Women	< 75	1999–2018	−0.10 (−0.50; 0.30)	-	-	−0.10 (−0.50; 0.30)
			0.606			0.606
	≥ 75	1999–2013	−0.23 (−0.62; 0.16)	2013–2018	−4.14 (−5.98; −2.26)	−1.27 (−1.80; −0.74)
			0.225		<0.001 *	<0.001 *

## Data Availability

Data were downloaded from the Instituto Nacional de Estadística (INE) website at https://www.ine.es (accessed on 27 January 2021).

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
