# Peer review of "Mortality Trends Related to Bladder Cancer in Spain, 1999–2018"

_jcm, 2022, doi:10.3390/jcm11040930_

Round 1

Reviewer 1 Report

Overall, this manuscript is well-written. There are a few comments. 

There are not enough discussions regarding why BC mortality has not changed in under 75 years women. Further discussion is needed.

The authors discussed on only tobacco as a risk of BC. Other chemical factors are also needed to discuss.

Page 3 L 143: Some words may be missed.

Author Response

Dear colleague, first of all thank you for your time and your interesting comments. 

1.- We have updated the discussion about why BC mortality has not decreased in <75 women, probably the main cause of this is the low mortality rates on this group. To help understand this, we have also added new tables in the results section:

4.3. Demographic variations, impact according to sex

According to the data analysed, premature mortality has not decreased in the women’s group. The mortality rate in this group is very low (age adjusted mortality rate 1-1.1 per 100,000 inhabitants), which makes demonstrating a decrease in mortality is very complicated, in contrast to the situation with men or women over the age of 75. Other conditioning factors, such as increased incidence in this group or social and health aspects that confer a worse prognosis must also be considered.

As women present with more advanced disease and have lower survival rates, female sex is considered an independent risk factor for mortality due to BC[54] and women with BC are considered to be at excess risk of mortality compared to men.[55–57] This can be explained by a delay in diagnosis; for example, symptoms such as microhematuria or irritative voiding symptoms may be misidentified as benign pathology (usually urinary infections) and so delay diagnosis of BC.[57]

On the other hand, changes in lifestyle and environmental exposure might increase BC incidence in women. An increase in the incidence rate of BC related to occupational exposure has been observed in women as compared to men, which may contribute to the fact that there has not been a decrease in the mortality rate in younger women does not decrease.[50] The increase in female smokers in the 1980s and 1990s (especially in young women), could be responsible for the rise in incidence and the consequent absence of a  decline in mortality.[7][9] If the incidence continues to increase, it is possible that in the next few years there will be an increase in mortality in these groups.

In the literature, a decrease in mortality has been described in European women,[24] but a stratified analysis by age allows us to affirm that premature mortality in women has not decreased, and this is probably the group that benefits the most from specific interventions.

New studies are needed in order to analyse whether delay in diagnosis is a factor and whether there have been changes in levels of exposure to risk factors based on gender, as well as to plan specific campaigns for early diagnosis and prevention of smoking in the population group of women under the age of 75.

2.- We have updated the discussion about the evolution in temporal exposure to other risk factors of bladder cancer:

4.1. Exposure to risk factors

The introduction of the Anti-Smoking Law (Law 28/2005) and the subsequent amendment (Law 42/2010) and the consequent decrease in smoking in Spain could explain the improvement in mortality figures. According to the data analysed, a change in mortality trends due to BC can be observed in men and women aged over 75 at 2 and 3 years respectively following enactment of the Law. Clearly, the variable latency period between exposure to tobacco, subsequent development of BC and mortality from the disease makes it difficult to assess the direct impact. In addition, this study was not designed to evaluate the impact of the Anti-Smoking Law on changes in smoking habits.

It is difficult to know the true impact of the Law for the Prevention of Risk at Work, and also of the different subsequent modifications to the Law, as these have not been evaluated. Additionally, it is difficult to attribute occupational exposure to carcinogens as the sole cause of BC.[3],[10] Although the incidence of BC related to occupational exposure increased in the decade 2000-2010, mortality risk declined for both sexes.[50] Furthermore, a reduction in risk over time has been observed in several occupations, such as drivers and mechanics.[50,51]

Other risk factors, such as infection by Schistosoma haematobium[1], are rare in Spain, and therefore these are not considered to have influenced the significant variation in the number of deaths.

3.- The sentence on page 3, L143 has been removed, as it was a sentence entered by mistake when adapting the manuscript to the journal's standards.

Kind regards, 

Reviewer 2 Report

The objective of this study is to analyze bladder cancer (BC) mortality trends between 1999 and 2018 in Spain. The results indicate that over the last 20 years there has been a significant decrease in the BC mortality rate, except in women under 75 years of age. Despite this temporal trend of decreasing mortality, BC continues to have a high impact on YPLL, mainly in men. The topic of this study is of clinical significance. However, the manuscript is lack of innovation and failed to provide instructive comments.

Major Problems:

Please further analyze the data in this study. For example, the authors mentioned “Health services: diagnosis and treatment” in Discussion section and how clinical diagnosis and treatment affect the mortality are expected to be studied.

Author Response

Dear colleague, first of all thank you for your time and your interesting comments. 

1.- We have updated the discussion about why BC mortality has not decreased in <75 women, risk factors for bladder cancer and the diagnosis and treatment section.

4.2. Health services: diagnosis and treatment

BC is considered an avoidable cause of death, both through primary prevention and also through early detection and treatment and the improvement of treatment and medical health.[42]

BC prognosis is stage and grade dependent. Delay in diagnosis and treatment does affect overall survival. BC diagnosis is initially through symptoms, and in this context primary care training is very important, along with advances in radiological tests (and access to these).

Based on these findings, we consider that a reorientation of health systems towards prevention, as well as the establishment of rapid diagnostic and treatment circuits, could be expected to contribute to the reduction of mortality rates due to BC and the disease’s impact on YPLL, without excessive additional costs.

The risk of recurrence is high for non-muscle-invasive BC patients, hence the use of biomarkers, cystoscopy surveillance and intravesical treatments can reduce the progression and thus mortality.[52]

Additionally, new treatments like PD-1/PD-L1 inhibitors have been shown to decrease mortality in patients with metastatic BC. However, their impact on the mortality trends analysed here should not be considered significant, as recommendations for their use are relatively recent[1] and more time is probably needed to observe changes in this area.

Moreover, the extension of the indications for radical cystectomy to elderly patients[20],[21] and the progressive incorporation of neoadjuvant chemotherapy[13,14] into treatment protocols in Spain could have contributed to the decrease in mortality observed in the trend analysis, particularly in the group aged 75 and over. For this reason, it is important to continue implementing bladder-sparring treatments, systemic treatments and to perform cystectomy in fit patients regardless of their age.[53] All of these contribute to reducing bladder cancer-specific mortality.   

Round 2

Reviewer 2 Report

The objective of this study is to analyze bladder cancer (BC) mortality trends between 1999 and 2018 in Spain. The results indicate that over the last 20 years there has been a significant decrease in the BC mortality rate, except in women under 75 years of age. Despite this temporal trend of decreasing mortality, BC continues to have a high impact on YPLL, mainly in men. The topic of this study is of clinical significance and the authors had carefully revised the manuscript according to the comments. I would praise all efforts the authors took in this study. However, the innovation and significance of the current results are insufficient. Further and deeper analysis of the current data are expected in the future work.

Major Problems:

  1. The authors had further discussed the results according to the comments. The mortality of BC was affected by various factors. Although the authors thoroughly discussed this topic in the Discussion section, the analysis of the current data did not show it. Further and deeper analysis of the data are expected in the future work

Author Response

  1. The authors had further discussed the results according to the comments. The mortality of BC was affected by various factors. Although the authors thoroughly discussed this topic in the Discussion section, the analysis of the current data did not show it. Further and deeper analysis of the data are expected in the future work. 

Thank you for your comment. We take the reviewer’s point, and have tried to assess and explain and the observed trends in line with the evidence of known risk factors in the literature, but in effect these explanations are not based on associations analysed in this paper.

A major limitation of this type of study is the absence of individual variables that can be associated with the observed trends by means of multivariate models using designs that allow causality to be established, and by extension allow the causes of the trends to be explored in depth, both at the level of data analysis and in the discussion, as the reviewer suggests.

This limitation is a common feature of time trend studies of both mortality and hospital admissions carried out in Spain, as the data sources do not incorporate individual variables, except for age and sex. The following are some examples of this type of design carried out in Spain with this limitation: Melchor et al. 2015; Orozco-Beltrán et al. 2017; Gomez et al. 2018; Verdu-Soriano et al. 2021; Carratalá-Munuera et al. 2021; Hervella et al. 2021; Orozco-Beltrán et al. 2021a; Orozco-Beltrán et al. 2021b.

We appreciate the reviewer’s comment, and we agree with him. However, we believe that this type of studies (with their limitations) are useful to gain an overview of the epidemiological situation of bladder cancer in Spain, and complement studies that make associations with risk factors.

It is to be hoped that in time we will have more complete sources of health information that include individual variables to overcome this limitation.

References

Melchor I, Nolasco A, Moncho J, Quesada JA, Pereyra-Zamora P, García-Senchermés C, Tamayo-Fonseca N, Martínez-Andreu P, Valero S, Salinas M. Trends in mortality due to motor vehicle traffic accident injuries between 1987 and 2011 in a Spanish region (Comunitat Valenciana). Accid Anal Prev. 2015 Apr;77:21-8. doi: 10.1016/j.aap.2015.01.023. Epub 2015 Feb 6. PMID: 25667203.

Orozco-Beltrán D, Sánchez E, Garrido A, Quesada JA, Carratalá-Munuera MC, Gil-Guillén VF. Trends in Mortality From Diabetes Mellitus in Spain: 1998-2013. Rev Esp Cardiol (Engl Ed). 2017 Jun;70(6):433-443. English, Spanish. doi: 10.1016/j.rec.2016.09.022. Epub 2016 Nov 5. PMID: 27825716.

Gómez-Martínez L, Orozco-Beltrán D, Quesada JA, Bertomeu-González V, Gil-Guillén VF, López-Pineda A, Carratalá-Munuera C. Trends in Premature Mortality Due to Heart Failure by Autonomous Community in Spain: 1999 to 2013. Rev Esp Cardiol (Engl Ed). 2018 Jul;71(7):531-537. English, Spanish. doi: 10.1016/j.rec.2017.09.026. Epub 2018 Jan 10. PMID: 29331563.

Verdú-Soriano J, Berenguer-Pérez M, Quesada JA. Trends in mortality due to pressure ulcers in Spain, over the period 1999-2016. J Tissue Viability. 2021 May;30(2):147-154. doi: 10.1016/j.jtv.2021.03.007. Epub 2021 Apr 3. PMID: 33836918.

Carratalá-Munuera C, Pilco JDR, Orozco-Beltrán D, Compañ A, Quesada JA, Nouni-García R, Gil-Guillén VF, García-Ortíz L, López-Pineda A. Hospitalization Trends for Acute Appendicitis in Spain, 1998 to 2017. Int J Environ Res Public Health. 2021 Dec 2;18(23):12718. doi: 10.3390/ijerph182312718. PMID: 34886447; PMCID: PMC8656947.

Hervella MI, Carratalá-Munuera C, Orozco-Beltrán D, López-Pineda A, Bertomeu-González V, Gil-Guillén VF, Pascual R, Quesada JA. Trends in premature mortality due to ischemic heart disease in Spain from 1998 to 2018. Rev Esp Cardiol (Engl Ed). 2021 Oct;74(10):838-845. English, Spanish. doi: 10.1016/j.rec.2020.09.034. Epub 2021 Jan 2. PMID: 33402321.

Orozco-Beltrán D, Arriero-Marin JM, Carratalá-Munuera C, Soler-Cataluña JJ, Lopez-Pineda A, Gil-Guillén VF, Quesada JA. Trends in Hospital Admissions for Chronic Obstructive Pulmonary Disease in Men and Women in Spain, 1998 to 2018. J Clin Med. 2021 Apr 6;10(7):1529. doi: 10.3390/jcm10071529. PMID: 33917437; PMCID: PMC8038653.

Orozco-Beltrán D, Guillen-Mollá A, Cebrián-Cuenca AM, Navarro-Pérez J, Gil-Guillén VF, Quesada JA, Pomares-Gómez FJ, Lopez-Pineda A, Carratalá-Munuera C. Hospital admissions trends for severe hypoglycemia in diabetes patients in Spain, 2005 to 2015. Diabetes Res Clin Pract. 2021 Jan;171:108565. doi: 10.1016/j.diabres.2020.108565. Epub 2020 Nov 23. PMID: 33242511.